# Individual and Work-Related Predictors of Exhaustion in East and West Germany

**DOI:** 10.3390/ijerph191811533

**Published:** 2022-09-13

**Authors:** Lisa Braunheim, Daniëlle Otten, Christoph Kasinger, Elmar Brähler, Manfred E. Beutel

**Affiliations:** 1Department of Psychosomatic Medicine and Psychotherapy, University Medical Center of the Johannes Gutenberg University Mainz, Untere Zahlbacher Str. 8, 55131 Mainz, Germany; 2Department of Psychiatry and Psychotherapy, University of Leipzig Medical Center, Semmelweis Str. 10, 04103 Leipzig, Germany

**Keywords:** exhaustion, technostress, burnout, information and communication technologies, East/West Germany

## Abstract

Chronic exhaustion is a consequence of detrimental working conditions and demands, as well as inadequate coping techniques, potentially resulting in burnout. Previous research has studied occupational environment and individual factors as predictors of exhaustion. Although these differ between former East and West German states, the regional distinction regarding exhaustion has been neglected. To fill this gap, we used the Copenhagen Burnout Inventory in a representative German sample from 2014 to assess the burnout symptom exhaustion. Estimating ordinary least squares regressions, important burnout predictors were compared between the former East and West German states. Regional differences concerning occupational environments were related to the associations between individual factors, situational aspects of technostress and exhaustion. Associations between individual factors (e.g., female sex, lower working hours, age, partnership status, and household income) and exhaustion were stronger in East Germany, whereas technostress (strain of internet use, number of e-mails during leisure time, and social pressure to be constantly available) was more strongly associated with exhaustion in West Germany. Despite lower financial gratification and a higher social pressure to be constantly available in the East, West Germans were more afflicted by exhaustion. Individual factors and technostress should thus be considered when focusing on job-related mental health issues.

## 1. Introduction

Information and communication technologies (ICTs) have reached a pervasive level in the labor domain. Conventional workday frames have become blurred as many organizational tasks are now independent of time and distance [1]. The growing intrusion of occupational aspects into leisure time might worsen the balance between the two domains. With less time to recover from everyday work-related stress, one might burn out [2] due to “excessive demands on energy, strength, or resources” [3]. Aside from work-related aspects such as ICT demands, individual factors play an important role for exhaustion or subsequent burnout [4]. The effort–reward imbalance model constitutes an example, in which work-related and individual factors both foster exhaustion through high demands and little gratification [5].

As a major public health concern, exhaustion increases behavioral health risk factors [6,7] and in turn the morbidity risk [8]. Malakh-Pines [9] defined three types of exhaustion: emotional, mental, and physical exhaustion. Emotional exhaustion can be described as feeling emotionally overextended and a perceived absence of emotional resources [10]. Mental exhaustion goes along with negative attitudes towards one’s self, one’s own life, or other people [9]. Physical exhaustion is associated with low energy, weakness, and weariness [9]. Overall, exhaustion is one of the symptoms of burnout [11] which is correlated with low mental and general health [12]. By the first definition of burnout, Freudenberger [3] listed frequent headaches, gastrointestinal problems, sleep trouble, and shortness of breath as its physical signs. In recent research, assessing burnout and differentiating it from exhaustion turned out to be problematic [8]. Van Dam [8] stated that clinical burnout could not be measured using the common questionnaires, as they overestimated the burnout prevalence due to the participants′ short-term stress, neglecting the duration of the symptoms. 

Previous research has shown the occupational environment in East Germany differs from West Germany, even after the reunification: compared to West Germany, East Germans′ jobs are more often shaped by Tayloristic work practices of monotonous production procedures [13]. This may be a remnant of the strong primary and secondary sectors of the economy of the German Democratic Republic (GDR), as opposed to the industrialized West as well as the Federal Republic of Germany (FRG) [14]. Therefore, the occupational environment may be associated with the general psychological health differences that were found between the two regions over the past 30 years [15]. However, despite their different work environments, exhaustion has not been compared between the former eastern and western states of Germany. Thus, this paper aims at adding knowledge to this lack of research. Regional differences of the occupational environment in the past and present are highlighted to determine their association with exhaustion predictors that may be related to an existing or impending case of burnout. As a result, the importance of focusing on the occupational environments within clustered regions is presented. The concepts exhaustion, burnout, technostress, and constant availability are introduced in the next steps. Additionally, the different occupational environments in East and West Germany are described. 

## 2. Theoretical Background

### 2.1. The Definition and Identification of Exhaustion and Burnout

Originally, the concept of burnout was related to people working in human service jobs who became “unable to cope with this continual emotional stress” [16] (p. 16). Kristensen et al. [12] have extended exhaustion as a key component of burnout to anyone. This stems from the definition of Schaufeli and Greenglass [17], who see emotionally demanding jobs as sources of physical, emotional, and mental exhaustion. Kristensen et al. [12] thus developed and validated the Copenhagen Burnout Inventory (CBI) and its subgroup on personal burnout, which asks for the intensity of one’s physical and psychological exhaustion unrelated to a certain domain. Still, they found its correlation with the other subgroup of work-related burnout (*r* = 0.72, *p* < 0.001) to be very high. 

For a long time, the International Classification of Diseases (ICD), ICD-10, merely mentioned burnout to be a “state of vital exhaustion” [18]. Only in the upcoming ICD-11 is burnout treated as an illness (QD85) under “Factors influencing health status or contact with health services” [11]. It is defined as a consequence of chronic work stress and characterized by three dimensions: (1) exhaustion; (2) mental distance from the job, negativism or cynicism towards it; and (3) feeling ineffective or lacking accomplishment [11]. Thus, as chronic work stress increases the risk for burnout [8,11], many questionnaires that try to assess burnout, such as the CBI [12] or the more commonly used Maslach Burnout Inventory [19], are not sufficient due to their lack of assessment of the symptoms’ time span. However, the duration of the feelings of exhaustion is crucial for the emergence of clinical burnout [8]. People suffering from clinical burnout oftentimes even become accustomed to their stressful lives until they collapse; while they were trying to maintain high standards of task performance, they were not able to recover from stressful times anymore [8]. Because of this circumstance, we used the term exhaustion instead. The majority of previous research used the term burnout, although their scale did not ask for the duration of the symptoms. Moreover, exhaustion can indicate more phenomena than only burnout, as multiple physical or mental health issues are related to it [20]. Since Kristensen’s et al. [12] burnout concept does not assess the duration of the symptoms, the term exhaustion is more appropriate. 

A meta-analysis by Shoman et al. [4] showed that situational and work-related (e.g., job demands, interpersonal relationships) as well as individual factors (e.g., personality traits, job attitudes), work–individual (conflicts, enrichment), and finally perceived intermediate work consequences (stress, satisfaction) predicted exhaustion in employees. Further, the work–family balance or work–life balance have an important mediating role when it comes to the association between ICT demands (e.g., pressure to be constantly available, interruptions during work time, work overload) and exhaustion as a higher balance reduces the risks of exhaustion [2]. Comparing several analyses, the Cohen’s f2 effect sizes of the work–family conflict range from small (<0.02) to medium (<0.15) [4]. Using prospective data, the correlations between job insecurity as well as emotional demands regarding the job and later burnout symptoms are positive [21].

### 2.2. Constant Availability and Technostress as Modern Side Effects

ICT use can simplify many aspects of everyday life and work, as it is able to structure work in a different, more independent way [22]. However, an overload of ICT use can be problematic. Ragu-Nathan et al. [1] developed the conceptual framework of technostress creators. First, constant connectivity, technoinvasion, enables people to be contacted anywhere and at any time; many of them feel forced to respond. Second, techno-overload explains how it becomes more difficult for workers to handle several mobile communication tools simultaneously as internal and external information increases. Previous research has shown adverse health effects related to technostress creators. Information overload and communication demands related to ICT use in the private sphere predict perceived stress for the group within the ages of 50 and 85 [23]. Moreover, Misra and Stokols [24] proposed that information overload as a consequence of the increasing use of ICTs were deleterious to attentional capacities and well-being. Technoinvasion was found to have a mediating role regarding the effect of techno-overload on burnout [25], using the scale developed by Malakh-Pines [26]. ICT demands have a higher impact on exhaustion, work–family balance and job satisfaction, which outweighs its supporting aspects [2]. A meta-analysis by Berg-Beckhoff et al. [22] revealed associations between ICT use within occupational settings and stress, whereas intervention studies did not find this. ICT use and burnout are positively associated, especially within the groups of middle-aged workers between the age of 35 and 45. Therefore, ICT demands were considered work-related predictors of exhaustion in this paper.

### 2.3. Different Occupational Environments in East and West Germany

Becker et al. [13] focused on past and present working conditions and aspects in East and West Germany. They reported that East Germans more often reported mental strains related to work than West Germans, especially due to financial loss or interruptions during work hours. In their argumentation, due to lower levels of wages before reunification and the rather slow convergence to West German standards, the ratio of income and working hours was lower in East Germany, providing West Germans with a higher financial gratification for their labor. Furthermore, they reported that East Germans worked more often in on-call duty as well as shift duty, and thus, needed to be available during leisure time more frequently than West Germans. In general, working conditions were partially less favorable in the former eastern states of Germany, posing a potential health risk for its employed inhabitants. 

However, in the GDR, compared to the FRG, a different and regulatorily broader approach to worker protection and occupational health existed, though its realization varied [13]. East German employees partly suffered from the adaptation to less protective West German occupational health after reunification, as they were accustomed to this form of working socialization. 

Nevertheless, more West Germans call in sick because of mental health issues which is why Becker et al. [13] assume that East Germans tend to continue working while being sick, exhibiting so-called presenteeism. This stems from having suffered from pervasive layoffs after the reunification and an insufficient protection of advocacies [27]. Moreover, East Germans tend to deny being ill [27]. This raises the question whether different working conditions and related stresses in the former eastern and western states of Germany affect exhaustion. 

### 2.4. Aims of the Study

The purposes of this study were twofold: (1)To examine the associations of work-related and individual predictors with exhaustion.(2)To examine whether associations between work-related and individual predictors with exhaustion differed between East and West Germany.

## 3. Materials and Methods

### 3.1. Sample

Data were based on a German representative survey from 2014 by the University of Leipzig that was approved by its ethics committee (Az.: 063-14-10032014). It assessed sociodemographic aspects as well as physical and mental well-being. The commercial survey institute USUMA (Independent Service for Survey, Methods and Analysis) collected the data, using a multistage random-route technique. First, 258 randomly drawn nonoverlapping regions from the last political election register, covering urban and rural areas from all regions in Germany, were selected. Out of these, 4386 households were randomly drawn. Using a Kish selection grid, household members of at least 14 years of age who understood the German language were chosen. A total of *N* = 2527 participants and thus 54.8% of the selected persons were questioned face to face. All participants gave their informed consent. 

Because of the focus on perpetual job-related availability and ICT use during leisure time, we excluded participants without employment. We also omitted participants with missing values on the used items. This led to a final sample of *N* = 1065. The detailed characteristics of our sample can be found in Table 1.

### 3.2. Measures

#### 3.2.1. Exhaustion

We used the German version of the six validated items of the component personal burnout of the CBI [12] to assess exhaustion. The questionnaire contains questions on physical, mental, and emotional exhaustion. It was asked how frequently the participants felt or thought in a distinct way. The answers ranged from 1 “never/almost never” to 5 “always”. To see if the personal burnout subgroup was related to the work domain, we used the three dimensions stated by the WHO [11] in regards to burnout: (1) the level of exhaustion was measured by the items of the personal burnout subgroup; (2) for the distance from the job and negativism, we chose the participants’ satisfaction with the job; (3) to assess the feeling of ineffectiveness and lack of accomplishment, we estimated the validated German perceived stress scale (PSS-10) [28], translated from Cohen et al. [29]. The correlations between the CBI subgroup personal burnout and the other mentioned dimensions of burnout are shown in Table 2. They ranged from moderate (*r* between ±0.30 and ±0.49) to high (*r* ≥ ±0.50) and pointed to the expected directions: a higher exhaustion score was negatively correlated with a higher job satisfaction (*r* = −0.323; *p* < 0.001) and the second factor of the PSS-10 (*r* = −0.296; *p* < 0.001), which assesses perceived self-efficacy [30]. The correlation with the first factor of the PSS-10, which indicates perceived helplessness [30], was high and positive (*r* = 0.619; *p* < 0.001). Thus, it could be assumed that the personal burnout subgroup, which we labelled exhaustion due to its neglection of symptoms′ duration, adequately assessed work-related exhaustion.

#### 3.2.2. Communication Load

On an 8-point scale ranging from 0 to >100, the number of job-related e-mails received during work as well as leisure time was measured. As the answers had categorical ranges, they were transformed into continuous variables, using the means of the ranges. Additionally, the strain of the Internet use regarding the participants’ work was asked. Between 0 “never” and 4 “very often”, they answered how often they perceived themselves to be strained because of their Internet use related to their work. 

#### 3.2.3. Constant Availability

Three items based on the perceived norm scale by Fishbein and Ajzen [31] assessed the importance of being perpetually available within the occupational environment (e.g., “I feel social pressure in my work life to be constantly available”). On a 5-point Likert scale the answers ranged from 1 “does not apply at all” to 5 “fully applies”.

#### 3.2.4. Work–Life Balance

Syrek et al. [32] developed and validated a scale to measure work–life balance, which consisted of five items asking for satisfaction with participants’ balance between work and private life. The answers varied on a 5-point Likert scale between 1 “strongly disagree” and 5 “strongly agree”. With Cronbach’s α = 0.95 and α = 0.88 in their validation paper of the scale, a good internal consistency, along with a good construct validity, was indicated.

#### 3.2.5. Sociodemographic Aspects

In the cross-sectional survey, we compared participants living in former western and former eastern regions of Germany. Moreover, age, sex, and household income were dependent variables. Household income originally was a categorical item but was transformed into a quasi-metric one using the means of the ranges. As only employed participants were included in the analyses, we constructed a dummy variable, differing between full-time and part-time workers. Because of the mediating role of work–life balance regarding burnout, we implemented dummy variables for having a partner in the household and having children as well.

### 3.3. Analyses

All analyses were calculated with Rstudio (version 1.4.1106) and its packages lavaan, psych, and arsenal. Descriptive statistics showed the numbers, mean values, and standard deviations of the used variables in the two regions so that sociodemographic differences between them could be observed. To test the significance of the regional differences, χ^2^- and F-tests were used. Using an ordinary least squares regression (OLS regression), predictors of exhaustion in Germany were found. An interaction term between the region and the respective predictors identified significant differences of the predictors between East and West Germany. In the final step, two separate OLS regressions were estimated for East and West Germany. 

## 4. Results

### 4.1. Descriptive Results

Descriptive results can be found in Table 1. Only cases without missing values are presented. The exhaustion factor score was lower in East Germany compared to West Germany (W.: 29.667 vs. E.: 26.294, *p* < 0.01). West Germans worked part-time significantly more often (W.: 25.70% vs. E.: 18.30%, *p* < 0.05). Moreover, the household net income (W.: 2855.162 vs. E.: 2407.396, *p* < 0.001) and the number of e-mails during work time (W.: 5.169 vs. E.: 3.586, *p* < 0.05) ranked lower in the former eastern states of Germany. The social pressure to be constantly available within the occupational environment was significantly higher in the East (W.: 33.182 vs. E.: 39.701, *p* < 0.01).

In Table 3, the exhaustion factor means for specific demographic subgroups, also differentiated between Easterners and Westerners, are presented. Men had lower mean values of the exhaustion score (male: 25.811 vs. female: 32.553, *p* < 0.001), while East German men were least concerned with this symptom (mean = 20.543). The differences within the subgroups household (W.: partner: 28.456 vs. no partner: 30.911, *p* < 0.05) and children (W.: yes: 28.877 vs. no: 31.504, *p* < 0.05) were only significant in West Germany. In the East, both having a partner in the household or not (E.: partner: 24.469 vs. no Partner: 28.576, *p* = 0.129), and having children (E.: yes: 26.259 vs. no: 27.105, *p* = 0.768) did not show a significant difference compared to the reference group. Contrary to that, in the West, participants without a partner in the household or with children reported significantly higher exhaustion scores.

### 4.2. Predictors of Exhaustion in Germany

Table 4 shows the regression results for Germany. With an adjusted R^2^ of 0.253, the model was able to explain 25.3% of the variance of exhaustion. East Germans were significantly less afflicted by it (std. β = −0.17, std. *p* < 0.05). Women (std. β = 0.25, std. *p* < 0.001) and also part-time workers (std. β = 0.15, std. *p* < 0.05), compared to those working full-time, had a higher exhaustion score. As expected, an increasing age was associated with increased exhaustion values (std. β = 0.16, std. *p* < 0.001). Being without a partner in the same household was positively related to exhaustion as well (std. β = 0.19, std. *p* < 0.01). Moreover, having children was associated with increased symptoms of exhaustion (std. β = 0.17, std. *p* < 0.01). The tendency of the association between an increased household income and exhaustion was negative, despite being insignificant (std. β = −0.17, std. *p* = 0.079). The factor score of a higher work–life balance was significantly and negatively related to exhaustion and exhibited the most influential protective factor (std. β = −0.39, std. *p* < 0.001). Regarding the technostress variables, the extent of feeling strained because of one’s Internet use (std. β = 0.08, std. *p* < 0.01) and the social pressure to stay connected within the occupational environment (std. β = 0.10, std. *p* < 0.01) were positively associated with exhaustion. Neither receiving e-mails during work time (std. β = 0.05, std. *p* = 0.112), nor during leisure time (std. β = 0.02, std. *p* = 0.461) were related to exhaustion. 

### 4.3. Predictors in East and West Germany

Figure 1 shows the predictors of the exhaustion factor score, differentiating between East and West Germany. Adding an interaction term between the regions to the full model above, two of the predictors were significantly lower in the East: the number of e-mails received during leisure time (std. β = −0.142, std. *p* < 0.01, not displayed in figure) and the social pressure to be constantly available within the occupational environment (std. β = −0.054, std. *p* < 0.05, not displayed in figure). Because of the small sample size of East Germans, several of the predictors’ variances were higher. Thus, it is plausible that larger samples could indicate further significant differences between the two regions. 

Among the sociodemographic variables, several predictors were higher in the East, although the interaction term between the two regions was insignificant. For instance, the difference between men and women was larger in the East than in the West (E.: std. β = 0.184 vs. W.: std. β = 0.108). This can be seen in Table 3 as well; East German men had the lowest exhaustion scores. The coefficients of having a part-time job (W.: std. β = 0.158 vs. E.: std. β = 0.266), age (W.: std. β = 0.148 vs. E.: std. β = 0.242), and being without a partner in the same household (W.: std. β = 0.087 vs. E.: std. β = 0.161) were also larger in East Germany. The directions of the associations of household income differed between the two regions, with a negative relation in the West (std. β = −0.229, std. *p* < 0.05), and a positive one in the East (std. β = 0.179, std. *p* = 0.470).

Some variables hardly exhibited any regional difference. The coefficients of having children were almost the same in the two regions, whereas for East Germans, it was not significant (std. β = 0.060, std. *p* = 0.482). The negative association with work–life balance was higher in the East, though the difference was comparably small (E.: std. β = −0.412 vs. W.: −0.380).

All variables regarding technostress, with the exception of the number of e-mails during work time, showed higher coefficients in the former western states. Being strained because of one’s own Internet use had a lower and insignificant association in East Germany (E.: std. β = 0.049, std. *p* = 0.505 vs. W.: std. β = 0.093, std. *p* < 0.01). Although the coefficients were insignificant, the number of e-mails received during work time were more linked with the exhaustion score in East Germany (E.: std. β = 0.119 vs. W.: 0.046). However, both receiving e-mails during work and leisure time were positively related to exhaustion in West Germany (leisure time: std. β = 0.032, *p* = 0.299); in contrast to work hours, e-mails during leisure time were significantly linked to a decreased exhaustion in the East (std. β = −0.167, *p* < 0.05). Finally, social pressure was only significantly associated with exhaustion in the former western regions (W.: std. β = 0.113, *p* < 0.001 vs. E.: std. β = −0.011, *p* = 0.890).

## 5. Discussion

Living in East Germany, being male, working full-time, having no children, and having a partner in the household were significantly negatively associated with exhaustion and might thus buffer it. Therefore, individual factors present important protective aspects regarding exhaustion. Having no children and having a partner in the household might contribute to a better work–life balance, which is still the largest protective factor for exhaustion. The variables related to the participants’ technostress only partially reached significance. Being strained because of one’s Internet use and the social pressure to be constantly available were related to exhaustion. The number of e-mails during leisure time was negatively associated with exhaustion only in East Germany, whereas a positive tendency was present in the West. 

The coefficients of technostress indicators especially differed between East and West Germany. In the West, a higher number of e-mails received during leisure time and the social pressure to be constantly available indicated technostress, which was related to exhaustion. In the East, sociodemographic aspects such as sex, working hours, age, or the partnership status tended to have higher associations with exhaustion. Thus, we assume that West Germans are especially affected by the social pressure to be available at all times, which is applicable to Shoman’s et al. [4] situational and work-related factors, while in East Germany, individual factors play a more important role than the occupational environment. Due to the significantly higher exhaustion rates as well as the larger coefficients of technostress regarding exhaustion in the West, it is possible that burnout is also more prevalent there.

Because of the constant access, independent from the workplace, due to ICT use, many employees are perpetually available during their work and leisure time. The work–life balance might suffer from this condition. Thus, stress and eventually exhaustion might be the consequence of a prolonged period of poor work-life-balance [2]. Besides exhaustion, a mental distance from the job and the feelings of ineffectiveness related to one’s work are indicators of burnout [11]. We confirmed the correlation between exhaustion and a lower job satisfaction as well as perceived self-efficacy and perceived helplessness. Situational and work-related, as well as individual, or work–individual factors, and perceived intermediate work consequences predict burnout [4] and thus also foster exhaustion.

ICT use in the private sphere significantly predicts perceived stress [22] with age as a moderator [23] and leads to demands that benefit burnout, a detrimental work–family balance and a worse job satisfaction [2]. Techno-overload, as a part of technostress, harms well-being [24] and fosters burnout, while a smaller amount of technoinvasion buffers this effect [25]. We confirmed the association between ICT use and exhaustion for West Germany. 

Whether respondents were from East or West Germany was dependent on their current living situation. Because of the still remaining structural, socioeconomic, and historic borders, differences regarding the occupational environment were expected to primarily depend on the current location, while the location of socialization, which is important for the individual factors, could not be considered due to a lack of data. However, as the place of work might be in another location than the household, this aspect needs to be kept in mind. Researchers on this topic should thus conduct questionnaires with full information on the location of socialization as well as the place of work.

With different occupational environments in East and West Germany, several health-related forms of behavior as well as illness behaviors also differ between the two regions. West Germans call in sick more often [13], which might lead to the assumption that sickness presenteeism is more common in the East. Still, we found that work-related exhaustion was less common in the former eastern states of Germany, especially regarding men. Although our descriptive results show that East Germans exhibit a higher social pressure to be permanently available, which is consistent with the finding that they work on call more often [13], their exhaustion scores ranked lower. Possibly, their occupational environment has a protective impact on their health. Another explanation derives from the question about social pressure to be constantly available. Future research may need to differentiate between these forms. It is possible that the availability related to on-call duty has a different impact than the availability related to e-mails or calls with other intentions. Further, it should be asked if the participants have the freedom of choosing whether they act on the message or not, as in the case of on-call duty. Thus, the effect of the occupational environment could be broken down in a better way.

## 6. Conclusions

In conclusion, we could show that the predictors of exhaustion might differ on a regional level. Former West German states revealed higher exhaustion rates than formerly East German states. The predictors also partially varied between the two regions, with aspects of technostress exhibiting stronger associations with exhaustion in the West and the tendency of stronger relations between individual factors and exhaustion in the East. The current and past occupational environment of the two regions should thus be highlighted when looking at burnout, its indicators, such as exhaustion, and mediators. Because of the differing indicators and outcomes of exhaustion in East and West Germany, varying effects between urban or rural regions are possible as well. At any rate, the results of this study can explain why outcomes are not consistent for different countries or groups in past studies, which was shown in the meta-analyses on exhaustion referenced in this paper.

## 7. Limitations

Several limitations should be considered when interpreting the results of this study. First, due to the focus on the working population, some sociodemographic groups in the East were of rather small numbers. Replications could benefit from larger samples. Second, the questionnaire contained only the personal burnout questions of the CBI. The work-related burnout items might have been a better indicator for job-related exhaustion. Furthermore, the duration of the feelings of exhaustion was not specified. To determine if a participant is afflicted by clinical burnout, the participant should exhibit the feelings of exhaustion for a longer period of time [8], in addition to the mental distance from the job and the feeling of ineffectiveness and lacking accomplishments [11]. Third, we did not have information on the participants’ previous history of mental health problems. Using a longitudinal design could improve these limitations and consider causality of the predictors. Fourth, we were not able to assess in which fields the participants of this study were employed. It is said that people who work with clients, in the role of human caring, suffer from burnout more often; it was not clear how the occupational positions were distributed in East and West Germany.

## Figures and Tables

**Figure 1 ijerph-19-11533-f001:**
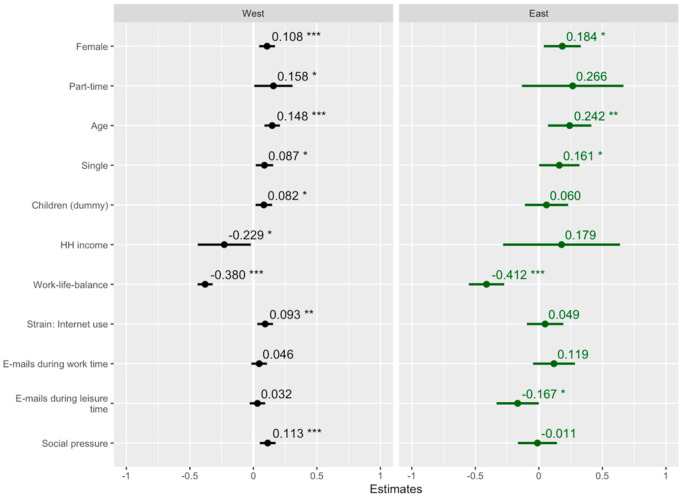
Predictors of exhaustion in East and West Germany. * *p* < 0.05, ** *p* < 0.01, *** *p* < 0.001. Standardized β coefficients, using two ordinary least squares regressions differed by region, are presented. East Germany: *N* = 169. West Germany: *N* = 896.

**Table 1 ijerph-19-11533-t001:** Study participants from East and West Germany.

	Total (*n* = 1065)			West (*n* = 896)			East (*n* = 169)			
Variable	N	%/Mean	SD	N	%/Mean	SD	N	%/Mean	SD	Significance Test ^1^
**Sociodemographic factors**										
Sex (women) ^2^	535	50.20%		448	50%		87	51.50%		χ^2^ = 0.072
Work hours (part-time) ^2^	261	24.50%		230	25.70%		31	18.30%		χ^2^ = 3.739 *
Household (no partner) ^2^	521	48.90%		446	49.80%		75	44.40%		χ^2^ = 1.449
Children (yes) ^2^	708	66.50%		590	65.80%		118	69.80%		χ^2^ = 0.837
Age	1065	42.650	11.369	896	42.542	11.392	169	43.219	11.259	F = 0.503
Household income (EUR)	1065	2784.108	1164.631	896	2855.162	1173.319	169	2407.396	1042.375	F = 21.42 ***
**Psychological factors**										
Exhaustion	1065	29.131	19.803	896	29.667	19.973	169	26.294	18.674	F = 4.136 **
Work–life balance	1065	70.043	21.894	896	69.847	21.905	169	71.082	21.873	F = 0.452
**ICT use**										
Strain: Internet use	1065	0.585	0.945	896	0.571	0.938	169	0.657	0.982	F = 1.16
E-mails during work time	1065	4.917	10.038	896	5.169	10.608	169	3.586	6.053	F = 3.543 *
E-mails during leisure time	1065	1.458	5.181	896	1.519	5.548	169	1.136	2.427	F = 0.776
Social pressure	1065	34.217	30.392	896	33.182	30.056	169	39.701	31.646	F = 6.576 **

^1^ * *p* < 0.05, ** *p* < 0.01, *** *p* < 0.001. ^2^ Only one category is presented for dummy variables, which is indicated in brackets.

**Table 2 ijerph-19-11533-t002:** Pearson′s product-moment correlations of dimensions of burnout with exhaustion factor score.

Item	*t*	*r*	*p*
Satisfaction with job	−11.732	−0.323	<0.001
Perceived self-efficacy	−10.669	−0.296	<0.001
Perceived helplessness	27.087	0.619	<0.001

**Table 3 ijerph-19-11533-t003:** Means of the exhaustion factor scores per group.

	Total			West			East		
	Mean	SD	*p* ^3^	Mean	SD	*p* ^3^	Mean	SD	*p* ^3^
**Region**			<0.05						
West	29.785	19.801							
East	26.543	19.105							
**Sex**			<0.001			<0.001			<0.001
Male	25.811	18.779		26.769	19.055		20.543	16.292	
Female	32.553	20.043		32.742	20.088		31.654	19.894	
**Household**			<0.05			<0.05			0.129
Partner	27.761	18.883		28.456	19.156		24.469	17.247	
No partner	30.550	20.340		30.911	20.281		28.576	20.650	
**Children**			<0.05			<0.05			0.768
Yes	30.802	20.108		28.877	19.553		26.259	18.986	
No	28.443	19.473		31.504	20.178		27.105	19.469	

^3^*t*-tests were performed according to regions, sex, household, and children.

**Table 4 ijerph-19-11533-t004:** Ordinary least squares regression: predictors of exhaustion.

	Factor Score Exhaustion				
Predictors	Estimates	std. β	std. CI	*p* ^5^	std. *p* ^5^
(Intercept)	52.35 ***	−0.10	−0.36–0.17	**<0.001**	0.479
East	−3.47 *	−0.17	−0.32–−0.03	**0.020**	**0.020**
Female	4.96 ***	0.25	0.14–0.36	**<0.001**	**<0.001**
Part-time (ref = full-time)	2.95 *	0.15	0.01–0.29	**0.037**	**0.033**
Age	0.28 ***	0.16	0.10–0.22	**<0.001**	**<0.001**
Household (ref = partner)	3.79 **	0.19	0.07–0.32	**0.003**	**0.003**
Children (ref = no children)	3.49 **	0.17	0.05–0.30	**0.006**	**0.007**
Household income (log)	−2.54	−0.17	−0.37–0.02	0.070	0.079
Work–life balance	−0.35 ***	−0.39	−0.44–−0.33	**<0.001**	**<0.001**
Strain: Internet use	1.66 **	0.08	0.02–0.14	**0.006**	**0.006**
E-mails during work time	0.09	0.05	−0.01–0.10	0.112	0.112
E-mails during leisure time	0.08	0.02	−0.03–0.08	0.462	0.461
Social pressure	0.06 **	0.10	0.04–0.15	**0.001**	**0.001**
Observations	1065				
R^2^/R^2^ adjusted	0.262/0.253				

* *p* < 0.05, ** *p* < 0.01, *** *p* < 0.001. ^5^ Significant *p* values (*p* < 0.05) are indicated in bold font.

## Data Availability

The data presented in this study are available on request from the corresponding author. The data are not publicly available due to licenses.

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
