# Peer review of "Individual and Work-Related Predictors of Exhaustion in East and West Germany"

_ijerph, 2022, doi:10.3390/ijerph191811533_

Round 1
Reviewer 1 Report
The flat battery of the West: The predictors of exhaustion in East 2 and West Germany.
The paper is very well-written and has a good flow. The subject is also very interesting and scientifically sound.
I point out my concerns here:
1. I guess the title of the article is a bit confusing to represent the research. I suggest changing to another title to have a better congruence with the content.
2. The conclusion is so short and does not cover all the aspects of the article.
Author Response
Dear reviewer,
thank you very much for taking the time to assess our work. We have attempted to revise the manuscript to include your constructive comments. Below you can find the answers marked with square bullets.
- I guess the title of the article is a bit confusing to represent the research. I suggest changing to another title to have a better congruence with the content.
- We changed the title to ‘Individual and work-related predictors of exhaustion in East and West Germany’ to account for the individual and work-related predictors we tested in the analyses.
- The conclusion is so short and does not cover all the aspects of the article.
- Thank you for pointing this out. We extended the conclusion to account for all the aspects of the manuscript, e.g. the difference in exhaustion rates between former Easternand Wester German states and the main predictors for exhaustion in both regions.
Best regards,
the authors
Reviewer 2 Report
Thank you for submitting this paper. The paper would benefit from re-organization and clarification.
Points for consideration:
How do you define exhaustion? This needs to be immediately defined as soon as you mention it in the introduction. It can be a brief definition – one or two lines (because you elaborate on this concept further down in the paper).
Introduction section - You mention that there are different work environments between East and West Germany. Please provide some examples. This provides a rationale into why research looking into these differences is needed.
You state “chronic work stress is the cause of burnout” – please change this to “chronic work stress increases the risk for burnout” and please provide reference(s) for this.
The authors mention – “Comparing several analyses, the effects of work-family-conflict are ranging from small to medium [4].” The authors do not indicate what small or medium effects are, please clarify. Please do this in other instances.
The authors state – “A meta-analysis by Shoman et al. [4] showed that situational and work-related as well as individual factors, work-individual, and finally perceived intermediate work consequences predict exhaustion in employees.” Please clarify this statement.
What do the authors mean when they state, sample points, in the Methods section?
Are those participants who were excluded from your study different in any way from those who remained in the study? Could you do simple statistics to compare (ex. sociodemographics comparisons) and determine whether there may be bias?
The authors state in the Results - "East Germans worked full-time significantly more often." Please present in brackets the numbers related to this finding. Please do this throughout the paper - the results need to be clearly presented.
Please proof-read the paper; suggestion to clarify and re-organize throughout.
Author Response
Dear Reviewer,
thank you very much for taking the time to assess our work. We appreciate your instructive comments and tried to include them in our revised manuscript. The answers to your comments can be found below marked by the square bullets.
- How do you define exhaustion? This needs to be immediately defined as soon as you mention it in the introduction. It can be a brief definition – one or two lines (because you elaborate on this concept further down in the paper).
- We added a commonly used definition from Malakh-Pines to the introduction. It describes three forms of exhaustion (emotional, mental, physical) that are also accounted for in the Copenhagen Burnout Inventory’s subgroup personal burnout that we assessed.
- Introduction section - You mention that there are different work environments between East and West Germany. Please provide some examples. This provides a rationale into why research looking into these differences is needed.
- We added examples here as suggested (ll. 61-5). Additional examples can be found in chapter 2.3.
- You state “chronic work stress is the cause of burnout” – please change this to “chronic work stress increases the risk for burnout” and please provide reference(s) for this.
- Thank you for pointing this out. We changed the working as suggested and added the references.
- The authors mention – “Comparing several analyses, the effects of work-family-conflict are ranging from small to medium [4].” The authors do not indicate what small or medium effects are, please clarify. Please do this in other instances.
- We clarified what was meant by small (Cohen’s f2 < 0.02) and medium (Cohen’s f2 < 0.15) effect sizes. Further, we added another clarification in ll. 230-1 to classify Pearson’s r values.
- The authors state – “A meta-analysis by Shoman et al. [4] showed that situational and work-related as well as individual factors, work-individual, and finally perceived intermediate work consequences predict exhaustion in employees.” Please clarify this statement.
- We added explanations right after the words that needed clarification. As previously, the sentence afterwards was meant to elaborate on the different factors, we deleted it to prevent repetition.
- What do the authors mean when they state, sample points, in the Methods section?
- With sample points we mean regions. In thus study, 258 non-overlapping regions were randomly selected. Within these regions, households were randomly selected and a target person within each household was selected using the Kish-Selection.Grid. We have clarified that in the methods section.
- Are those participants who were excluded from your study different in any way from those who remained in the study? Could you do simple statistics to compare (ex. sociodemographics comparisons) and determine whether there may be bias?
- As we excluded those who did not work in either full-time or part-time, as expected, there are significantly more women and a significantly lower household income in the group of excluded participants. The other sociodemographic aspects did not differ on a significant level.
- The authors state in the Results - "East Germans worked full-time significantly more often." Please present in brackets the numbers related to this finding. Please do this throughout the paper - the results need to be clearly presented.
- We added the results, using brackets, throughout the paper.
- Please proof-read the paper; suggestion to clarify and re-organize throughout.
-
- We had the paper proof-read and clarified several passages. Many style and grammar mistakes were corrected.
Best regards,
The authors
Reviewer 3 Report
The work entitled “The flat battery of the West: The predictors of exhaustion in East 2 and West Germany” contains new scientific knowledge and covers a relevant topic. However, I have some comments that should be considered before publication.
- In the participants section, did authors check for previous history of mental health problems. If not, this should be mentioned in the limitations section.
With regards to the variables. I do not know if authors should consider variables like partner/partner or children/no children as it i sor if it would be interesting to analyze the number children and see if that affects the results or if being married, divorced os single (for instance is affecting the results). It is likely that it is not the same having 1 or 3 children.
I think that it would be interesting to indicate the criteria for being parto f East and West Germany, as some regions could be located very close to what would be considered as west or east respectively,
Also, authors should be cautious when establishing conclusions, as the total amount of participants from the East regions is limited. For instance, only31 of them were part time workers or 75 had no partner. The conclusions drawn from that limited amount of participants could be totally bias.
Author Response
Dear reviewer,
than you very much for taking the time to assess our work as well as your constructive comments. We hae attempted to incorporate them in the revised version. Below, you can find the answers to your comments marked by the square bullets.
- In the participants section, did authors check for previous history of mental health problems. If not, this should be mentioned in the limitations section.
- We did not have information on the history of mental health, which is why we added this point to our limitations.
- With regards to the variables. I do not know if authors should consider variables like partner/partner or children/no children as it i sor if it would be interesting to analyze the number children and see if that affects the results or if being married, divorced os single (for instance is affecting the results). It is likely that it is not the same having 1 or 3 children.
- Regarding family status: We assume a relation between exhaustion and living together with a partner since the partner is able to offer, besides emotional support, help regarding household chores and caretaking. Estimating a model with a categorical differentiation (single, married, divorced/married but separated, widowed), we only found a significant difference between single and married which is very similar to our original result. The number of observations is also too small in the East (31 divorced, 7 widowed) to include this differentiation in our models.
- Regarding the number of children: In the East, only 24 participants reported to have more than 1 child; this number of observations is too small to differentiate between the number of children in our analyses.
- I think that it would be interesting to indicate the criteria for being parto f East and West Germany, as some regions could be located very close to what would be considered as west or east respectively,
- We added to the discussion that work-related factors should thus be accounted for as structural, socioeconomic, and historic boarders can still be found. However, we do not know about the place of socialization or place of work. Future questionnaires should conduct these information.
- Also, authors should be cautious when establishing conclusions, as the total amount of participants from the East regions is limited. For instance, only31 of them were part time workers or 75 had no partner. The conclusions drawn from that limited amount of participants could be totally bias.
- We agree with that, which is why we added this to the limitations.
Best regards,
The authors
Round 2
Reviewer 3 Report
Authors have addressed all my previous comments. I have no further suggestions.